# α-Lactalbumin mRNA-LNP Evokes an Anti-Tumor Effect Combined with Surgery in Triple-Negative Breast Cancer

**DOI:** 10.3390/pharmaceutics16070940

**Published:** 2024-07-14

**Authors:** Yun-Ru He, Heng Xia, Peng Yun, Yuandong Xu, Winson M. J. Ma, Ze-Xiu Xiao, Gao-Feng Zha

**Affiliations:** 1Guangdong Provincial Key Laboratory of Digestive Cancer Research, The Seventh Affiliated Hospital, Sun Yat-sen University, No. 628, Zhenyuan Road, Guangming District, Shenzhen 518107, China; heyr23@mail.sysu.edu.cn (Y.-R.H.); xiah9@mail.sysu.edu.cn (H.X.); xuyuandong@sysush.com (Y.X.); 2Scientific Research Center, The Seventh Affiliated Hospital, Sun Yat-sen University, No. 628, Zhenyuan Road, Guangming District, Shenzhen 518107, China; 3Department of Endocrinology, The Seventh Affiliated Hospital, Sun Yat-sen University, No. 628, Zhenyuan Road, Guangming District, Shenzhen 518107, China; yunp3@mail.sysu.edu.cn; 4Shenzhen Institute of Advanced Technology, 1068 Xueyuan Avenue, Shenzhen University Town, Shenzhen 518055, China; mj.ma@siat.ac.cn; 5Drug Discovery Center, Shenzhen MagicRNA Biotech, Shenzhen 518107, China

**Keywords:** triple-negative breast cancer, mRNA vaccine, LNP, immune response, oncology–immunotherapy

## Abstract

Triple-negative breast cancer (TNBC) has been considered a huge clinical unmet need due to its aggressive progression and highly frequent metastasis. mRNA therapeutics supply a potential and versatile immunotherapy of oncology treatment. Here, we developed α-lactalbumin mRNA-lipid nanoparticles (α-LNP) as a potential therapeutical strategy for TNBC. The α-LNP induced the specific IgG antibodies and activated IFN γ-secreting-T cells in vivo. Additionally, the safety of α-LNP also had been demonstrated in vivo. When vaccinated prior to tumor implantation, α-LNP showed a preventive effect against 4T1 tumor growth and extended the survival of the tumor model by activating the memory immune responses. Furthermore, α-LNP administration in combination with surgical removal of neoplasm effectively inhibited the progression and metastasis in the TNBC model. Taken together, our results indicate that the α-LNP vaccine is a promising novel treatment for both therapeutics and prophylactics in TNBC.

## 1. Introduction

Breast cancer (BC) is the leading cause of female cancer-related deaths and has taken over lung cancer as the most commonly diagnosed cancer all around the world, according to the World Health Organization’s reporting since 2021, with increasing rates in the later years [1,2]. Among different types of cancer, triple-negative breast cancer (TNBC) is a BC subtype with highly invasive metastasis, aggressive recurrence and poor prognosis, which lacks effective targets for clinical treatment [3,4]. Nowadays, patients commonly undergo resection surgery, radiation therapy and/or conventional chemotherapy. However, some patients may develop chemotherapy resistance or discontinue due to serious adverse drug reactions (ADR) [5]. The targeted therapies may be on the horizon with poly-ADP-ribose polymerase (PARP) inhibitors and immune checkpoint, PD-1/PD-L1 inhibitors, which have limited improvement in the overall survival of TNBC patients [6,7,8]. Clinical outcomes for TNBC still remain unsatisfactory. The median overall survival for patients with metastatic TNBC is approximately eighteen months, much lower compared to the HR-positive and HER2-enriched BC where survival may exceed eight years [4,9]. Thus, the optimal treatment strategy for TNBC patients still remains a clinical unmet need.

α-Lactalbumin is a specific differentiation protein only expressed in mammary epithelial cells during lactation and over-expressed in the majority of human breast cancer [10,11,12,13]. The protein primarily exists in TNBCs and these cell populations are positively correlated with the ‘triple-negative’ state and unfavorable prognosis of BC patients. Moreover, some studies indicate that women have a substantial proinflammatory T cell repertoire responding to human recombinant α-lactalbumin [10]. And the Vincent research found that immunoreactivity against α-lactalbumin provides effective protection and therapy against TNBC in MMTV-neu and MMTV-PyVT transgenic mice and also against 4T1 transplantable tumors in BALB/c mice [10,14]. The recombinant α-lactalbumin protein can activate tumor-infiltrating lymphocytes (TILs) and show a predominance of both CD4^+^ and CD8^+^ T cells [10]. A Phase I clinical trial of adjuvant therapy with an α-lactalbumin vaccine in patients with non-metastatic TNBC at high risk of recurrence has been initiated recently [15]. Since α-lactalbumin is conditionally expressed only during lactation, vaccination-induced prophylaxis occurs without any detectable inflammation in normal non-lactating breast tissue. Thus, α-lactalbumin may serve as a very promising target for TNBC.

With the effective fighting of Coronavirus disease 2019 (COVID-19), mRNA vaccination has recently sparked tremendous interest in laboratory and clinical research [16,17,18]. Technological advances have optimized mRNA-based vaccine stability, delivery carrier and mRNA modification, and mRNA vaccines have been viewed as a promising option for not only infectious disease prevention but also cancer therapy [19,20]. Encouraging results from early clinical trials of mRNA vaccines have been obtained, and the research targets melanoma, glioblastoma, non-small-cell lung cancer lung cancer and so on [20]. In a recent phase I clinical trial, personalized mRNA neoantigen vaccines in combination with anti-PD-L1 and conventional chemotherapy induced substantial T cell activity and demonstrated an increased recurrence-free survival in pancreatic ductal adenocarcinoma (PDAC) patients with resected tumors. A phase 2 study is currently recruiting based on this breakthrough clinical data [21]. With the advantages of personalized preparations, low manufacturing costs and effectively scalable production, mRNA vaccines are reaching their potential as a future crucial strategy for cancer treatment [22,23].

In this work, we proposed a potential clinical strategy for TNBC therapeutics. Based on the effective and safe delivery system, the mRNA vaccine encoding α-lactalbumin showed remarkably enhanced T cell response and special antibody production and thereby a preventive effect on the 4T1-implanted model. Moreover, the combination of the TNBC mRNA vaccine with surgical removement achieved a prolonged survival on the established 4T1-implantation model with an excellent inhibitory effect on lung metastasis, revealing robust antitumor immunity generated by our mRNA cancer vaccine.

## 2. Materials and Methods

Cell Culture

The 4T1 mouse breast cancer cell (CAT#CL-0007) and HEK293T cell lines (CAT#CL-0005) were purchased from Procell Life Science & Technology Co., Ltd. (Wuhan, China). 4T1, and HEK293T cells were cultured in RPMI 1640 (Thermo Fisher Scientific, Waltham, MA, USA), with 10% fetal bovine serum (FBS; Gibco, Grand Island, NY, USA) and 100 U/mL penicillin–streptomycin (Gibco, Cat #151401-22). All cells were grown at 37 °C in 5% CO_2_ conditions. And all cells were negative in mycoplasma detection.

Design and Production of mRNAs

The mRNA expression plasmids (pIVT), which is reconstructed based on pUC57 (GenScript, CAT#1176), contained T7 promoter, the optimized protein-coding sequence, 5′ and 3′ UTRs from the COVID-19 vaccine mRNA-1273 (Moderna, Cambridge, MA, USA) and an A (30) +GCAUAUGACU+A (80) poly (A) tail (BioNTech, Mainz, Germany). The optimized α-Lactalbumin CDS sequence (GC content from 46.8% to 57.3%) was synthesized and inserted between 5′ and 3′ UTR by seamless cloning technology (Beyotime, Cat #D7010M) with primers:

α-L-F: 5′-GGCGCCGCCACCATGCACTTCGTGCCTCTGTTCCTGGTG-3′

α-L-R: 5′-GGCTCCAGCTCATCAGGGCTTCTCGCATCTCCACTGCT-3′

pIVT-F: 5′-TGATGAGCTGGAGCCTCGGTGGCCT-3′

pIVT-R: 5′-CATGGTGGCGGCGCCGGGGTCTTA-3′

Before transcription in vitro, plasmids were extracted (Axygen, Tewksbury, MA, USA, Cat #UE-MD-P-25) and linearized with BsaI restriction endonuclease (Novoprotein, Suzhou, China, Cat #RE036) prior to in vitro transcription. mRNAs were produced following the instruction of T7 RNA polymerase kit (Novoprotein, Cat #E121). The UTP was 100% instead by one-methylpseudouridine (m1Ψ)-5′-triphosphate (TriLink, San Diego, CA, USA) to produce modified mRNA, then enzymatic capped with vaccinia capping enzyme and 2′-O-methyltransferase (Novoprotein, Cat #M072 and M062). mRNA was purified with RNA Clean Magnetic Beads (Vazyme, Nanjing, China, Cat #N412). RNA Clean Beads were added to the original RNA solution and mixed well. Incubation for 5 min at room temperature made the RNA binding on magnetic beads. Then, the sample was placed in a magnetic rack for 5 min, and then we carefully removed the supernatant. Keeping the tube in the magnetic rack, we added 500 μL 80% ethanol to wash magnetic beads, incubated these for 30 s, carefully removed the supernatant and repeated the wash step once. We eluted and collected RNA from the magnetic rack by adding a suitable volume of nuclease-free H_2_O. The concentration of mRNA was measured using NanoDrop One (Thermo) and diluted to 1.0 ug/μL and RNA quality was assessed by using Agilent 5200 Fragment Analyzer (Agilent Technologies, Palo Alto, CA, USA) and AGE (agarose gel electrophoresis). mRNA was stored at −80 °C before use.

Synthesis and Characterization of mRNA-LNPs

The ionizable lipid (A1A3) has been thoroughly detailed in our earlier article [24]. Lipids and mRNA were mixed in a microfluidic device (MicroNano, Shanghai, China) at a 3:1 volume ratio (N/P ratio of lipids/mRNA was 5:1). The lipids included the synthesized ionizable lipid, DSPC (Avanti Polar Lipids, Alabaster, AL, USA), cholesterol (Avanti) and DMG-PEG 2000 (Avanti), with a molar ratio of 44.2:9.9:44.2:1.7, which are solubilized by ethanol. The mRNA was prepared in 200 mM NaOAc buffer (pH 5.0). Then, the mRNA-LNP mixture was diluted with PBS in a 100,000 MWCO cassette (Merck, Darmstadt, Germany, CAT#UFC9100) at 4 °C for 30 min three times, centrifuged at 1200× *g* for 30 min and then stored at 4 °C before use.

The diameter and polydispersity index (PDI) of mRNA-LNPs were measured by Zetasizer Nano (Malvern Instruments, West Midlands, UK). The concentration and encapsulation rate of mRNA were measured by Quant-iT RiboGreen RNA Assay Kit (Invitrogen), the detecting method following the manufacturer’s protocol.

mRNA-LNPs Transfection and Expression

HEK293T cell lines were cultured in RPMI 1640 medium (Gibco) supplemented with 10% fetal bovine serum (BI) and 1% penicillin–streptomycin (Gibco, Cat #151401-22). The 24-well plate was inoculated with 2.5 × 10^5^ HEK293T cells for each well 24 h before the transfection of 0.5 μg mRNA per well using Liposomal Transfection Reagent (YEASEN, Cat #40802ES03). The cultured medium supernatant was harvested at 24 h (α-lactalbumin is predicted to be a 16.13 kDa protein containing a secreted signal peptide “MHFVPLFLVCILSLPAFQA”) and concentrated by 3 KD ultrafiltration (Millipore, Cat #UFC500396). The condensed medium was diluted by sodium carbonate solution (pH 9.6) and was coated overnight in the 96-well plate (Nest, Wuxi, China) at 4 °C, and then blocked with 5% bovine serum albumin solution (Sigma-Aldrich, St. Louis, MO, USA). The plates were washed three times and incubated with rabbit anti-mouse α-lactalbumin antibody (NOVUS, Cat #NBP3-03561) for 3 h at 37 °C. The plates were washed three times by PBST and incubated with 50 μL of 3,3′,5,5′-tetramethylbenzidine substrate (Beyotime, Shanghai, China, P0209) for 10–20 min. Next, 50 μL of 2 M phosphoric acid at 37 °C and OD 450 absorbance was measured by a microplate reader (SYNERGY H1).

For Western blotting, in addition to the medium supernatant concentrated above, the cell in each well (24-well plates) was lysed by 200 μL 2×SDS-loading buffer (Solarbio, Beijing, China, Cat #P1018), respectively, and then boiled at 98 °C for 5 min. Rabbit anti-mouse α-lactalbumin antibody (NOVUS, Cat #NBP3-03561) and HRP-linked anti-rabbit IgG antibody (CST, Cat #7076) were used to detect in this study.

Cryo-electron Microscopy

The LNP sample in solution was spread on a grid (Xinxingbairu, Beijing, China, CAT#T11032, #T11012), which formed a very thin liquid. Then, the grid was put into ethane for flash freezing. After flash freezing, the sample was transformed into amorphous ice. Cryo-EM samples are prepared by the Vitrobot Mark IV (Thermo Fisher) and then observed by TEM (Kiros G4, Einstein).

Animal Experiments

An amount of 5/6-week-old female Balb/C mice were purchased from the GemPharmatech company (Nanjing, China) and all mice were kept in a pathogen-free environment at Shenzhen Bay Laboratory (Shenzhen, China). The animal experiments were conducted in accordance with protocols approved by the Medical Animal Experiment Center’s Experimental Animal Ethics Committee of Shenzhen Bay Laboratory (Assurance Number: AEZGF202201). All animal experiments were carried out in strict conformity with the Chinese Laboratory Animal Regulations and Laboratory Animal Environment and Housing Facility Requirements.

For the immunotherapy model, 2 × 10^5^ 4T1 cells were injected subcutaneously in the right flank of the Balb/C mice. The mice were first immunized with 5 μg mRNA-LNP after 10 days and then four more administrations every five days were performed. For the prophylactic model, the mice were intramuscularly immunized with 3 μg α-LNP or LUC-mRNA-LNP or non-RNA-LNP or PBS three times every 7 days. Then, 2 × 10^5^ 4T1 cells were injected subcutaneously in the right flank of the Balb/C mice. Tumor volume was measured with formulation of ½ × (length × width × height) every 2/3 days and the tumors were limited by a volume of 1500 cm^3^. The weight of mice was recorded every 3 days.

To evaluate α-LNP safety, the mice were immunized with 5 μg α-LNP three times every 5 days. And the organs (liver, lung, heart, kidney) and blood were collected 5 days after the last administration.

Enzyme-Linked Immunosorbent Assay (ELISA)

The antibody titer of serum was measured by ELISA. α-lactalbumin (Cloud-Clone Corp, Wuhan, China) was diluted in sodium carbonate solution (pH 9.6) and was coated (3 μg per well) overnight on the 96-well plate (Nest, USA) at 4 °C. The plates were then washed four times by PBS with 0.5% Tween-20 (PBST) and blocked with 5% bovine serum albumin solution (Sigma-Aldrich). The serum collected from immunized mice was diluted in a serial dilution of 1:1000, 1:10,000, 1:100,000 and 1:1,000,000. And then the diluted serum was added to the plates for 4 h at 37 °C. After washing the plates four times by PBST, it was incubated with 1:2000 HRP-linked anti-mouse IgG antibody (CST) for 1 h at 37 °C. The plates were washed four times and incubated with 50 μL of 3,3′,5,5′-tetramethylbenzidine substrate (Beyotime, Cat #P0209) for 10–20 min. An amount of 50 μL of 2 M phosphoric acid at 37 °C and OD450 absorbance was measured by a microplate reader (SYNERGY H1) and the finite titer of serum was determined as the highest dilution, of which absorbance reached to/over four times of PBS group.

The levels of IFN-gamma, TNF-alpha and IL-12P70 in serum were detected by an ELISA kit (Biolegend, Shenzhen, China) according to the manufacturer’s instructions.

BMDC Activation and Enzyme-Linked Immunosorbent Spot (ELISPOT)

The production of IFN-γ secreted by splenocytes, isolated 7 days after the last immunization, was measured by ELISPOT. Bone marrow-derived dendritic cells were collected from 6–8-week-old female Balb/C mice. Then, the immature BMDCs were stimulated in RPMI 1640 with 10% FBS, 20 ng/mL GM-CSF (Peprotech, Suzhou, China) and 20 ng/mL IL-4 (Peprotech). And the cells were half-renewed with the medium every 2 days and were harvested on day 6, and then the phenotypes of dendritic cells were detected by CD11c expression. When CD11c was expressed in over 70% of cells, the BMDCs were incubated with 1 μg mRNA-LNP in a 6-well plate (4 × 10^5^ cells per well) for 24 h. The splenocytes (2 × 10^5^ cells per well) from the immunized mice were harvested and co-incubated with the immunized BMDCs (5 × 10^4^ cells per well) at a ratio of 4:1 in the ELISPOT plate. The following procedure was performed according to the mouse IFN-gamma precoated ELISPOT Kit (Biolegend, Cat #2210005).

Flow Cytometry

The female BALB/c mice were immunized with mRNA-LNP three times and were sacrificed 7 days after the last administration. The cells in tumor-infiltrating lymph nodes were harvested from immunized mice or untreated mice and then blocked by anti-mouse CD16/32 (Biolegend) for 20 min at 4 °C. The splenocytes from immunized mice were harvested and suspended to single cells, with erythrocytes lysing by ACK buffer. The lymphocytes were then stained with the following: PE-conjugated anti-mouse CD45 (Biolegend), APC-conjugated anti-mouse CD3 (Biolegend), BV610-conjugated anti-mouse CD4 (Biolegend), FITC-conjugated anti-mouse CD8α (Biolegend), BV421-conjugated anti-mouse CD44 (Biolegend), Per-CY5.5-conjugated anti-mouse CD62L (Biolegend), PECY7-conjugated anti-mouse GL7 (Biolegend), BV421-conjugated anti-mouse B220 (Biolegend), PE-conjugated anti-mouse CD138 (Biolegend), BV610-conjugated anti-mouse CD11b (Biolegend), PE-conjugated anti-mouse CD11c (Biolegend), BV421-conjugated anti-mouse CCR7 (Biolegend), BV610-conjugated anti-mouse CD80 (Biolegend), BV421-conjugated anti-mouse MHCII (Biolegend), PE-conjugated anti-mouse CD206 (Biolegend) and PECY7-conjugated live–dead. Analysis was performed using a FACS Caliber flow cytometer (BD Biosciences, San Jose, CA, USA) and analyzed using cytexpert2.4 and flowjo10.8.1. The flow cytometry strategies are illustrated in Appendix A.

Immunofluorescence (IF) and Hematoxylin–Eosin (HE) Staining

The tumors were harvested and fixed in 4% paraformaldehyde for 24–48 h, and then dehydrated by ethanol and embedded in paraffin. The 4 μm of tumor tissue slides were dewaxed in xylene twice for 10 min, and subsequently were rehydrated in 100%, 90%, 80% and 70% ethanol and PBS buffer for 5 min. For immunofluorescence, slides were blocked with 5% goat serum in PBS buffer and then incubated with primary antibodies (1:400, CST CAT#9718) at 4 °C for 8–12 h and secondary antibodies at 37 °C for 1 h. The slides were washed with PBS before being mounted with Vectashield Antifade Mounting Medium with DAPI (Vectorlabs, Burlingame, CA, USA, H-1200). The slides were imaged by confocal microscopy. For HE staining, the tissue slides were stained by hematoxylin (Beyotime) for 10 min and followed by eosin (Beyotime) for 5 min. The images were captured by microscopy (Pannoramic MIDI) and analyzed by slide converter (3D HISTECH, Budapest, Hungary).

Statistical Analysis

A two-tailed Student’s t-test or a one-way analysis of variance (ANOVA) was performed when comparing two groups or more than two groups, respectively. The normality test was checked by the Shapiro–Wilk test and equal variance was checked by Levene’s test. Statistical analysis was performed using Microsoft Excel LTSC 2021 and Prism 9.0 (GraphPad). Data are expressed as means ± s.d. Difference was considered to be significant if *p* < 0.05 (* *p* < 0.05, ** *p* < 0.01, *** *p* < 0.001, unless otherwise indicated). The survival rates of the two groups were analyzed using a log-rank test and were considered statistically significant if *p* < 0.05.

## 3. Results

### 3.1. Characterization of mRNA-LNPs Encoding α-Lactalbumin

The modified mRNA vaccine, encoding α-lactalbumin, was encapsulated in LNPs. The α-lactalbumin mRNA was modified with N1-Methylpseudouridine-5′-Triphosphate(m1ψTP) instead of native UTP and then purified by RNA Clean Magnetic Bead. The modified mRNA molecules start with a 5′-cap, then move on to the 5′-UTRs, coding region and 3′ UTRs, and end with a poly A tail. As depicted in Figure 1A, the α-lactalbumin mRNA performed well with purity and integrity. We generated the α-lactalbumin mRNA-LNP(α-LNP) by mixing aqueous buffer and ethanol phase with a ratio of 3:1 through a micro-fluid mixer. The average diameter of the α-LNP was 83.13 ± 18.9 nm and the polydispersity index (PDI) showed as 0.10 ± 0.02. The mRNA encapsulation efficiency (EE%) of α-LNP was 92.82 ± 1.52% (Figure 1B), high-performance capillary electrophoresis results also showed the mRNA in good condition before and after encapsulation (Appendix A), demonstrated the good quality of α-LNP. As shown in Figure 1C,D, the integrated shape and the well-distributed state of α-LNP were evaluated by cryo-electron microscopy. After transfection of the α-LNP into cells, Western blots analysis revealed a clear band of 17 kD, which suits the proper protein size of α-lactalbumin, indicating that the α-LNP could be expressed α-lactalbumin well (Figure 1E). And α-lactalbumin also secreted in the culture supernatant from the α-LNP transfected cells, displayed the potential of humoral immunity initiation (Figure 1F). With the well-evaluated parameters of α-LNPs, we successfully formulated the α-lactalbumin mRNA-LNP, and it could be expressed into α-lactalbumin well in vitro.

### 3.2. α-LNPs Demonstrated Strong Immunogenicity and Safety Profile In Vivo

To examine the immunogenicity of α-LNP in vivo, we immunized female mice with α-LNP 3 times by intramuscular injection and then monitored the IgG titer of peripheral blood on Day 15 (Figure 2A). As shown in Figure 2B, α-LNP induced a thousand times of IgG production in mice compared to the PBS-treated control group, suggesting that α-LNP activated robust humoral immunity. Furthermore, enzyme-linked immunospot (ELISpot) analysis showed that the mice immunized with α-LNP stimulated splenocytes produced significantly higher levels of IFN-γ compared to the PBS group, and concanavalin A group, which is regarded as a positive control (Figure 2C–E). To evaluate the cytokines in the blood, we then detected the level of interferon-γ (IFN-γ), tumor necrosis factor-α (TNF-α) and interleukin-12 (IL-12) in peripheral blood. After three treatments of α-LNP with an interval of 5 days, the level of IFN-γ, TNF-α and IL-12 of experiment mice showed no obvious change in serum (Figure 2F). The aspartate transaminase (AST), blood urea nitrogen (BUN) and creatinine (CREA) showed insignificant increases in peripheral blood. Though the alanine aminotransferase (ALT) was lower than the normal levels, all these biochemical data were in the normal range (Appendix A). In addition, tissues of main organs from the experiment-grouped mice were isolated for histopathological examination, and there were no pathological changes between the vaccination and control groups (Appendix A). Our results revealed that the mRNA vaccine based on α-LNP has an adequate safety profile in vivo. Collectively, α-LNP is a promising TNBC mRNA vaccine candidate because of its strong immunogenicity and safety profile in vivo.

### 3.3. α-LNPs Displayed Prophylactic Effect on TNBC In Vivo

Vincent’s research found that the vaccine with recombinant α-lactalbumin showed a time-dependent effect in anti-tumor response in transplantable 4T1 tumors and could be a potential prophylactic vaccine candidate [10]. A 4T1 mammary carcinoma is a transplantable model and is considered to be a common TNBC model close to the TNBC progress and metastases in clinics [25]. As the α-LNP initiated significant humoral and cell immune responses previously (Figure 2B–E), we hypothesize that α-LNPs treatment would prevent the growth of TNBC. In the 4T1 xenograft mice model, we observed a significant inhibitory effect on tumor growth and a total extension of 11 survival days (Figure 3A–D) in the α-LNP-treated group of mice. All of the mice’s weights showed no obvious change after vaccination (Appendix A). And the LNPs in the different groups were confirmed in their size, PDI, and percentage of encapsulation (Appendix A). After 34 days inoculation with 4T1 tumor cells, the immune responses of experiment mice were roundly analyzed. The CD4^+^CD44^+^CD62L^+^ (memory) T cells and CD4^+^CD44^+^CD62L^−^ (effector) T cells (Figure 3E) were significantly increased in α-LNP treatment group mice. CD8^+^CD44^+^CD62L^+^ T cells (memory) and CD8^+^CD44^+^CD62L^−^ T cells (effector) obviously showed an increase in infiltrating lymph nodes (Figure 3E), suggesting that α-LNP developed an immunological memory response. CD80 represents the maturation [26], and CCR7 suggests the immigrated capability of antigen-presenting cells [27,28,29]. We found that CD80 and CCR7 increased on DCs from tumor-infiltrating lymph nodes (Figure 3F), suggesting that the α-LNP improves the LNDC maturation and boosts its migration ability. Based on our results, α-LNP initiated effective anti-tumor immunity and prevented the growth of the 4T1-transplantable TNBC tumor model.

### 3.4. Surgical Removement Combined with α-LNPs Inhibited Tumor Growth and Metastasis of TNBC

All TNBC is an aggressive BC subtype and surgical operation is the basic monotherapy, as most patients will proceed to surgery first in the clinic [30,31]. Considering this clinical practice, we monitored the effect of α-LNP by combining neoplasm surgical removement. As depicted in Figure 4A, we first constructed a 4T1 transplantable mice model and conducted the surgery to remove the 9/10 primary tumors, the rest of the tumors represent the residual or recurring tumors after surgery in clinical, and the model mice were intramuscularly administrated with α-LNPs. As shown in Figure 4B,D, all the mice in the control group hit the experiment limit (tumors grew to 1500 mm^3^) on Day 32, and the surgical group of mice reached the experiment endpoint on Day 35, demonstrating that the surgery showed little tumor-inhibitory effect. The surgical removement combined with α-LNP administration significantly restricted the tumor growth and extended survival time by 20 days, and displayed 1/8 complete remission, as shown in Figure 4C. Metastasis is the leading cause of mortality in TNBC patients, and nearly 36.9% of metastatic breast cancer is found as lung metastasis [32]. Moreover, the amount of lung metastasis tumors was obviously restrained in the combinational treatment group, demonstrating that the α-LNP suppressed the metastasis of TNBC (Figure 4E). The body weight showed no obvious change in each group (Appendix A). Taken together, the results suggested that surgery combined with α-LNP administration inhibited the TNBC growth and metastasis, and, notably, extended the survival time.

### 3.5. α-LNPs Inhibit TNBC by Effectively Activating Immune Profile

To clarify the activated immune profile of α-LNPs in TNBC models, we analyzed the dendritic cells in tumor-infiltrating lymph nodes. The flow cytometric results showed that α-LNP improved the expression of CD80 and CCR7 on DCs (Figure 5A). In addition, CD138^+^B220^−^ cell, representing the antibody-secreted plasmacyte, and germinal center B cell (GL7^+^B220^+^), which responded to the increasing serum antibody levels, increased in the spleen from α-LNPs-treated mice (Figure 5B). CD8^+^ T cells increased in the spleen from the α-LNPs-treated group of mice (Figure 5C) and CD4^+^ T cells showed no significant change (Appendix A). CD206 expression demonstrated the anti-inflammatory effect of macrophage and CD86 represented the pro-inflammatory effect of macrophage [33,34]; we found that CD206 significantly decreased while CD86 showed no change in the spleen from experiment-grouped mice (Figure 5D and Appendix A), indicating that α-LNPs facilitated the anti-tumor response of the tumor-burdened mice. Furthermore, we detected the immune environment in situ and the immunofluorescence (IF) results (Figure 5D) suggested that there were more CD8^+^ T cells infiltration and IFN-γ expression in the surgery combined α-LNPs treated group. In terms of a conclusion, α-LNPs activated the anti-tumor immuno-profile in lymph organs and effectively increased the CD8^+^T cell and increased CD8^+^ T cells infiltration in the tumor, thus suppressing the TNBC growth and metastasis.

## 4. Discussion

TNBC is the most malignant subtype of BC and has presented an unmet need in clinics due to its aggressive behavior and poor prognosis. Given the evidence supporting the effectiveness and safety of mRNA vaccines, mRNA treatment is poised to become one of the major pillars in cancer therapeutics development [23]. mRNA-based individualized neoantigen therapy mRNA-4157, prolonged recurrence-free survival in patients with resected high-risk melanoma, another neoantigen-based mRNA vaccine autogene cevumeran (BNT122) induces effective T cell response and inhibits recurrence of PDAC [21,35]. And epitope-specific vaccines have also experienced positive evaluation for the high efficiency of mRNA delivery in vivo and the discovery of tumor-associated antigens (TAA) and tumor-specific antigens (TSA). An RNA-LNP vaccine (BNT111), targeting four TAAs of melanoma, durably induces adoptive T-cell therapy and shows positive utility in phase II clinical of advanced melanoma patients [36]. Accumulating trials demonstrated that mRNA vaccines are reaching their potential as a future crucial strategy for cancer treatment. In this study, we develop the mRNA vaccine encoding α-lactalbumin(α-LNP), based on our previous research on delivery platforms for mRNA vaccines [24].

Local inflammation occurs at the site of injection of mRNA vaccines, with infiltration of APCs, like myeloid DCs, monocytes and macrophages [19]. As demonstrated, the mRNA vaccine is taken by the APCs, which is responsible for the translation of the encoded protein and antigen presentation in lymphoid tissue, after the uptaken and expression of targeted mRNA-LNP, expression of co-stimulatory markers such as CD80 and CD86 increased to further drive the adaptive cellular mediated immune response. In our study, α-LNP administration activated the DCs and stimulated splenocytes to produce substantial IFN-γ in vivo, which demonstrated strong immunogenicity. A previous study reported that the migrativity of LNDC upregulated also suggested the improvement of antigen-specific recognition and T cell priming [37]. We found that CCR7 increased on DCs from tumor injection site draining lymph node after the α-LNP administrated in vivo, demonstrating that α-LNP increased the recruitment of APCs to lymphoid tissue and further drove the adaptive immune response.

Studies indicate that activated innate immunity triggers adaptive immunity, including cellular responses and neutralizing antibody production [38]. And, according to our results, α-LNP activated the T cell repertoire and effectively induced the B cell repertoire including plasmacyte activation and anti-tumor antibodies production. Humoral immunity resists the tumor through direct neutralization and the antibody-dependent cellular cytotoxicity (ADCC) effect [39,40]. In our study, the significantly increased α-lactalbumin specific antibodies, GL7^+^ B cells and CD138^+^ plasmacytes have been observed in the α-LNP-vaccinated group of mice, we supposed that the anti-tumor efficiency might be partly due to the ADCC response. As for the cellular-mediated anti-tumor responses, the CD8^+^ T cells and IFN-γ expression show obvious tumor-infiltrated after the α-LNP administration. The immune response activation is always followed by the negative feedback to reduce effector T cell functions by recruitment of immunosuppressive regulatory T cells (Tregs) or myeloid anti-inflammatory macrophages (CD206 high expression), and anti-inflammatory-related cytokines secretion [41,42]. According to our result, the CD206^+^ macrophages in the spleen are much less in α-LNP immunized mice.

According to the previous studies, the tumor-burdened mice that underwent surgical removement displayed rapid recurrence [43], we also found that the mice undergoing tumor resection only to remove 4T1 showed recurrence, while the surgery combined with α-LNP administration significantly restricted the tumor growth and suppressed the lung metastasis of TNBC. The α-LNP immunization of the model mice at an earlier tumor growth period displayed an improved antitumor effect, which is similar to the result of Vincent [10]. Notwithstanding, the appropriate administration time of the treatment is another key query for our future work on TNBC.

Overall, we reported that α-LNP, an mRNA vaccine encoding α-lactalbumin, activates the specific anti-tumor responses including cellular and humoral immunity. In combination with basic surgery operation, the mRNA vaccine displays desirable therapeutic treatment on the murine TNBC model, which enables more strategy for TNBC that correlates with extended survival time and delayed recurrence.

## Figures and Tables

**Figure 1 pharmaceutics-16-00940-f001:**
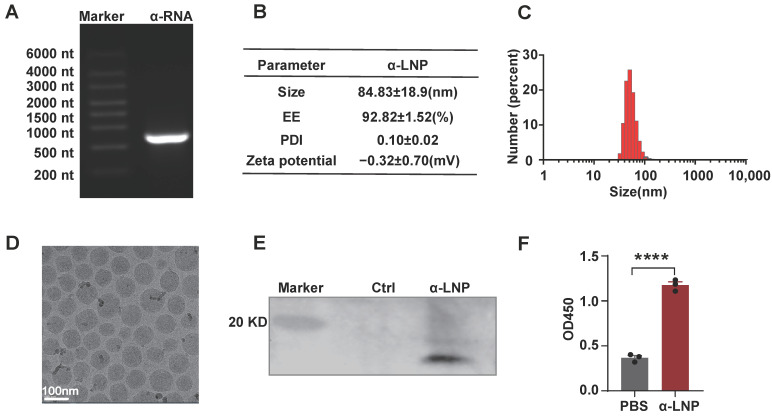
Characterization of mRNA-LNPs encoding α-lactalbumin and expression in vitro. (**A**) representative agarose electrophoresis image of α-lactalbumin mRNA. (**B**) Average size, PDI, zeta potential and encapsulation efficiency of α-LNPs. (**C**) Size distribution of α-lactalbumin mRNA-LNPs (α-LNPs). (**D**) The representative cryo-electron microscopy image of α-LNPs. Scale bar = 100 nm. (**E**) α-LNPs encapsulated with α-lactalbumin mRNA were transfected into 293T cell line, and the α-lactalbumin was detected by Western blotting. (**F**) The cultured medium was collected and concentrated by a 100 kD ultrafiltration device, and the supernatant was diluted with coating buffer and precoated on the 96–well plate for 24 h. After blocking with 5% BSA buffer and incubating with rabbit antimouse α-lactalbumin antibody (1:1000), the secreted α-lactalbumin expression in the medium was detected by ELISA. The quantitation data were represented with 3 experimental replicates, **** *p* < 0.0001, error bars represent mean ± SEM.

**Figure 2 pharmaceutics-16-00940-f002:**
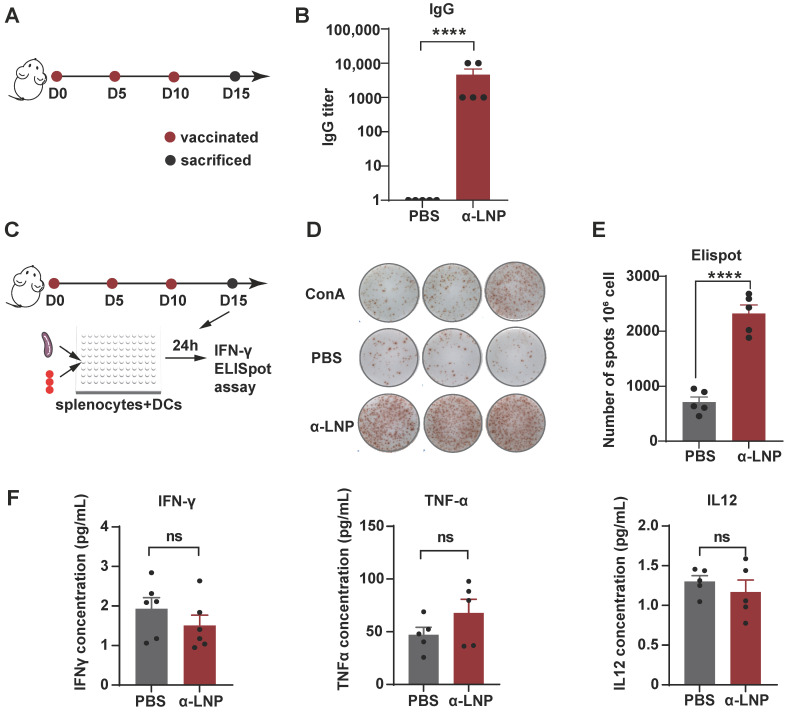
α-LNPs induce robust immunogenicity and safety profile in vivo. (**A**) Illustration of the intramuscular immunization timeline. After three times vaccination of α-LNPs, the mice were sacrificed 5 days after the last administration. *n* = 5. (**B**) The blood was collected after 3 immunizations, and the serum IgG titer was detected by ELISA, compared with control group (PBS administration), *n* = 5. The data are shown as the mean ± SEM. (**C**) Illustration of the activation and secretion of splenocytes. Enzyme-linked immunospot (ELISpot) analysis of IFN-*γ*-secreting splenocytes after co-cultured with BMDC which was stimulated with α-LNPs/PBS/ConA for 24 h, Concanavalin A (ConA) as the positive control. (**D**) The representative image of IFN-*γ* forming spots in ELISpot experiment. Each group was represented with 3 experimental replicates. (**E**) The quantitation data of ELISpot were represented with 5 biological replicates. *n* = 5. (**F**) The concentration of IFN-γ/TNF-α/IL-12 in serum among different groups. *n* = 5, **** *p* < 0.0001, error bars represent mean ± SEM. No sigificance (ns).

**Figure 3 pharmaceutics-16-00940-f003:**
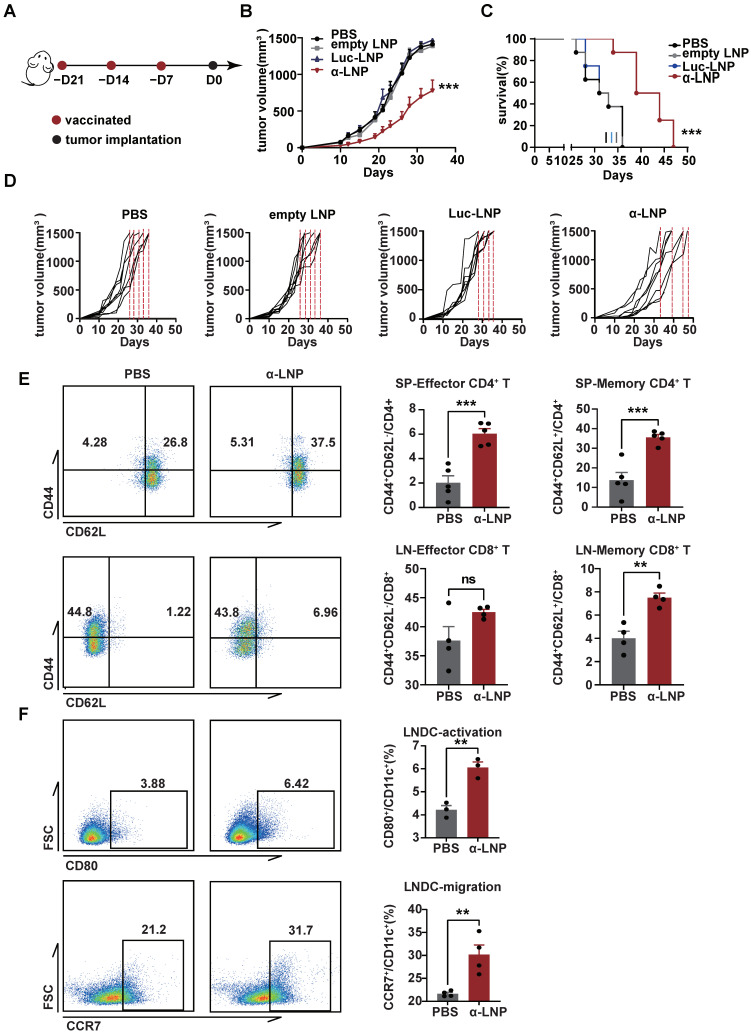
α-LNPs show prophylactic effect on triple-negative breast cancer in vivo. (**A**) Illustration of the vaccine immunization and tumor implantation timeline. After 3 vaccinations of α-LNPs, the 4T1 cells were implanted in the flank of female mice. *n* = 8. (**B**) The tumor growth in different groups of mice after vaccination. (**C**) The survival rate of mice. *n* = 8. (**D**) The tumor growth curve of each mouse in different groups. *n* = 8. (**E**) Representative flow cytometry plots (left) of memory CD4^+^ T cells and effector CD4^+^ T cells in spleen and statistical data of left figure. *n* = 4. Representative flow cytometry plots (left) of memory CD8^+^ T cells and effector CD8^+^ T cells in lymph nodes and statistical data of left figure. *n* = 5. (**F**) Representative flow cytometry plots of activation (CD80^+^CD11c^+^) and migration (CCR7^+^CD11c^+^) ability on DCs in lymph nodes. *n* = 5. Data are shown as means ± SEM, ** *p* < 0.01, *** *p* < 0.001. No sigificance (ns).

**Figure 4 pharmaceutics-16-00940-f004:**
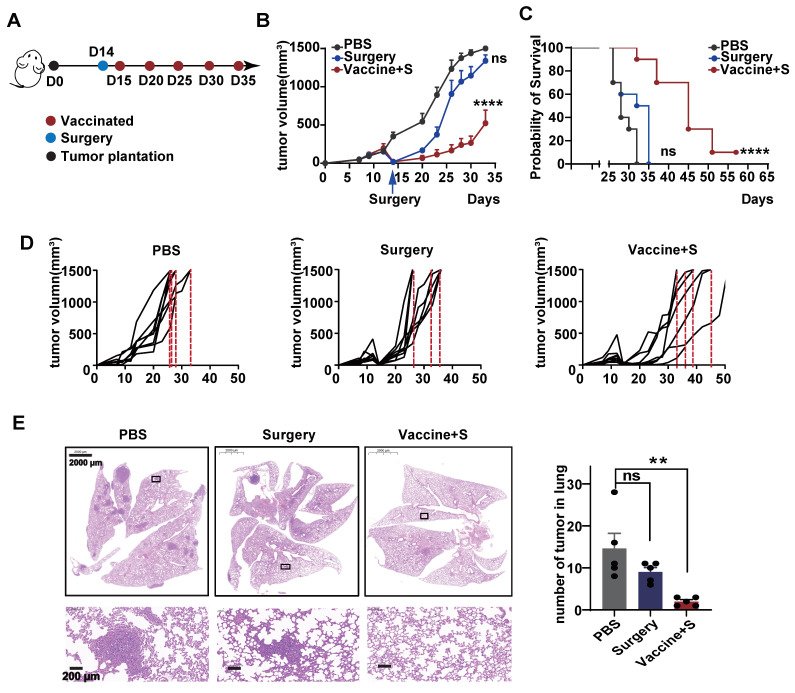
Surgery combined with α-LNPs inhibited tumor growth and metastasis in TNBC. (**A**) Illustration of the intramuscular therapeutical immunization timeline. After tumor implantation, the mice underwent surgery to remove 9/10 volume of the tumors and were vaccinated with 5 µg α-LNPs the next day. α-LNPs were immunized 5 times, with administration every 7 days, *n* = 8. (**B**) The tumor growth in different groups of mice after tumor implantation. (**C**) The survival rate of mice in different groups. (**D**) The tumor growth of each mouse in different groups. (**E**) The representative HE staining image of lung tissue in different groups. And statistical data are shown (right). *n* = 5. ** *p* < 0.01, **** *p* < 0.0001, error bars represent mean ± SEM. No sigificance (ns).

**Figure 5 pharmaceutics-16-00940-f005:**
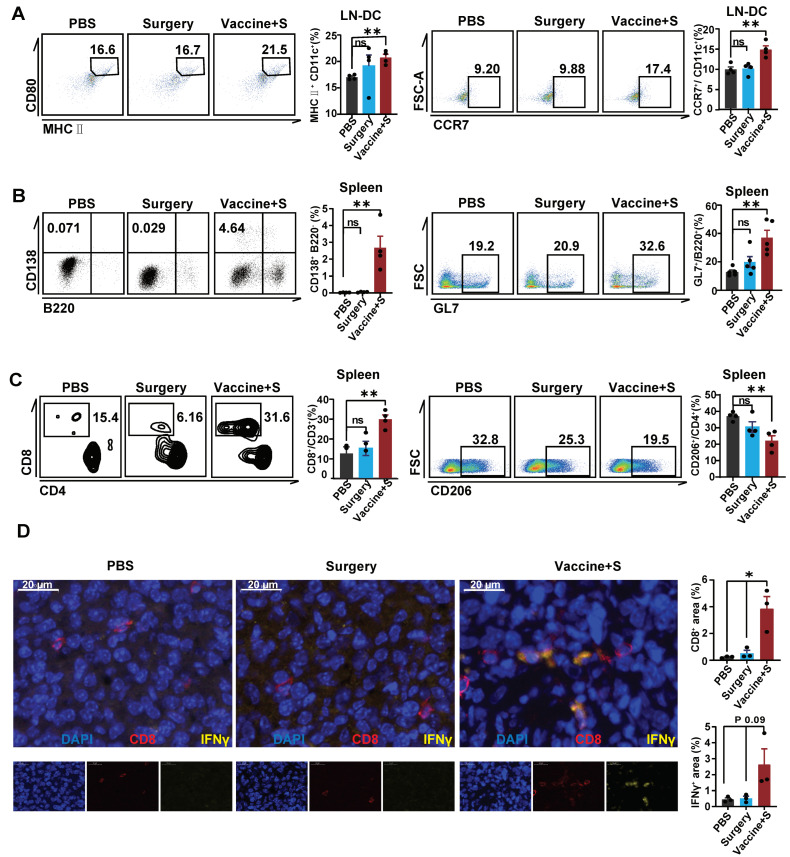
α-LNPs inhibit TNBC by effectively activating the immune profile. (**A**) Representative flow cytometry plots of DCs in lymph nodes. The flow cytometry was shown with CD80^+^ MHC II^+^ CD11c^+^ cells and migratory CCR7^+^ CD11c^+^ cells in lymph nodes. *n* = 4. (**B**) Representative flow cytometry plots of B cells in spleen. The flow cytometry results show GL7^+^ B220^−^ (plasmacytes cells) and GL7^+^ B220^+^ cells (GCB) in spleen, *n* = 4. (**C**) Representative flow cytometry plots of T cells in spleen (left). And statical data (right) shows CD4^+^ T cells and CD8^+^ T cells in spleen, *n* = 4. (**D**) Representative immunofluorescence staining in tumor tissue slides are shown, with the surgery combined vaccination group, surgery group and PBS group as control. *n* = 3 in each group. Scale bars = 20 μm. The quantification data of immunofluorescence staining are calculated following the formula of CD8^+^ area/total nuclear area or IFN γ^+^ area/total nuclear area. The mean positive area percentage of each slide was based on 5 visions. The quantitation data were represented with 3 replicates. * *p* < 0.05, ** *p* < 0.01. error bars represent mean ± SEM. No sigificance (ns).

## Data Availability

The original contributions presented in this study are included in the article and the Appendix A and further inquiries can be directed to the corresponding author/s.

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
