# Peer review of "α-Lactalbumin mRNA-LNP Evokes an Anti-Tumor Effect Combined with Surgery in Triple-Negative Breast Cancer"

_pharmaceutics, 2024, doi:10.3390/pharmaceutics16070940_

Round 1

Reviewer 1 Report

Comments and Suggestions for Authors

In this work, He and collaborators explore the therapeutic potential of a lipid nanoparticle (LNPs) mRNA-based vaccine to elicit T and B cell immunity to α-lactalbumin (α-LNP), a protein selectively induced in triple negative breast cancer cells or in normal mammary cells during lactation. The generation and physicochemical characterization of LNPs are clearly described and robust. The authors then characterize the efficacy of α-LPNs in a 4T1 tumor transplantation model. In both prophylactic and therapeutic regimens following surgical rejection, α-LNP showed promising improvements in enhancing the adaptive immune responses. The in vitro and in vivo experiments are generally well-conducted with appropriate controls, making this work solid and convincing. 

However, some minor and major concerns should be addressed before considering the work suitable for publication.

Minor Concerns:

1.      Line 274: It might be more appropriate to say “PBS-treated control group” or “unvaccinated mice” instead of “PBS group.”

2.      Quantification of ELISPOT: The quantification of ELISPOT in Figure 2E does not seem to correspond to the representative images shown in Figure 2D, where splenocytes from α-LNP-treated mice display at least ten times more spots compared to the control condition.

3.      Figure legend of Figure 2F: The figure legend for Figure 2F should be revised for clarity.

4.      Typo in figure 4A: There appears to be a typo in the experimental design depicted in Figure 4A (“plantation” should likely be “implantation”).

5.      Personal statements: In Line 354, avoid using personal statements like “to our satisfaction”. I recommend using a more scientific language instead.

6.      Statistical analysis: The authors used parametric tests such as one-way ANOVA and Student’s t-test, which assume a normal distribution of the data. Clarification is needed on whether the normality of the data was assessed before applying these tests. Information on the normality tests conducted (e.g., Shapiro-Wilk test) and any transformations applied to meet the assumptions of parametric testing would be beneficial. If normality cannot be assumed, non-parametric alternatives (e.g., Kruskal-Wallis test, Mann-Whitney U test) should be considered and discussed.

Major Concerns:

1.      Vaccination schedule rationale: The rationale behind the vaccination schedule shown in Figure 2A is unclear. Typically, a priming and boost regimen involves a first immunization followed by a recall after at least 14 days to allow time for the adaptive immune response to develop, in a similar way to what performed in data associated to figure 3.

2.      Assessment of humoral immunity: The authors evaluated the induction of specific antibodies after α-LNP vaccination (Figure 2A,B). However, the impact of humoral immunity in the antitumor preclinical evaluation has not been assessed either experimentally or in more details through discussion.

3.      Prophylactic potential and PD-1 therapy: The data describing the prophylactic and post-surgical rejection potential of α-LNP are encouraging, but the effect is not striking. 

TNBC is generally considered a "cold" tumor, characterized by a lower level of immune activity within the tumor microenvironment, which often makes them less responsive to immunotherapy treatments compared to "hot" tumors that have higher levels of immune infiltration and activity. The interesting advantage of vaccination might lie in the enhancement of intratumoral T cell immunity, which is often kept silent in the immunotolerant tumor environment by PD-L1:PD-1 interaction. The presence of a higher number of T cells, especially CD8+ T cells, is a favorable prognostic factor. These “hot” tumors, characterized by substantial T cell infiltration, are more responsive to checkpoint inhibitor therapy.

Assessing the tumor microenvironment for T cell infiltration could provide critical insights.

·      Combination with Checkpoint Inhibitors: While the vaccination alone shows promising results, it might significantly enhance the impact of the work to combine it with anti-PD-1 therapy. This combination could help to reveal whether the vaccinated mice, despite displaying moderate control of tumor growth on their own, exhibit a more responsive and "hotter" tumor microenvironment due to increased T cell infiltration.

·      Experimental verification: This hypothesis could be verified by evaluating the immune cell infiltration status using techniques such as flow cytometry or immunofluorescence histology, similar to the methods employed in Figure 5. Specifically, assessing PD-1 expression on infiltrating CD8+ T cells and other immune markers would be beneficial.

·      Clinical implications: If the combination therapy proves effective, it could represent a significant advancement in the treatment of TNBC, making the tumors more responsive to checkpoint inhibitors and potentially transforming the therapeutic landscape for this aggressive cancer subtype.

·      Additional studies: Future studies should consider investigating the synergistic effects of α-LNP vaccination and PD-1 blockade in preclinical models. This approach could elucidate the potential for combined immunotherapy regimens to induce robust anti-tumor immunity and improve clinical outcomes in TNBC patients.

Comments on the Quality of English Language

1.      English language: The manuscript could benefit from a thorough review to improve the overall English language quality.

2.      Sentence structure: Authors frequently start sentences with “and,” which is grammatically incorrect in formal writing.

Author Response

Comments 1: Line 274: It might be more appropriate to say “PBS-treated control group” or “unvaccinated mice” instead of “PBS group.”

Response 1: Thank you for pointing this out. We have modified the phrase to “PBS-treated control group”, as can be seen in Line 287 of revised manuscript.

(Line 287): “As shown in Figure. 2B, α-LNP induced thousand times of IgG production in mice com-pared to PBS-treated control group, suggesting that α-LNP activated robust humoral im-munity.”

Comments 2: Quantification of ELISPOT: The quantification of ELISPOT in Figure 2E does not seem to correspond to the representative images shown in Figure 2D, where splenocytes from α-LNP-treated mice display at least ten times more spots compared to the control condition.

Response 2: The quantification of ELISPOT in Figure 2E is based on the number and morphology of spots, which is recognized and presented by AID Elispot7.0. The reason why quantification of ELISPOT seems non-correspond to the representative images may be some spots in α-LNP-treated mice have too much positive signal to form the clear foci, which will not be counted as the specific positive spot, making the number of spots recognized by camera less than by eyes.

please check the complete figures of ELISPOT in the attachment file.

Comments 3: Figure legend of Figure 2F: The figure legend for Figure 2F should be revised for clarity.

Response 3: Thank you for pointing this out. We have revised the legend of Figure 2F and deleted “(F)The serum level of ALT/AST/BUN/CR in different groups”, as it can be seen in Line 315-316.

Comments 4: Typo in figure 4A: There appears to be a typo in the experimental design depicted in Figure 4A (“plantation” should likely be “implantation”).

Response 4: Thank you for pointing this out. We have corrected “plantation” to “implantation” in figure 4A of revised manuscript.

Comments 5: Personal statements: In Line 354, avoid using personal statements like “to our satisfaction”. I recommend using a more scientific language instead.

Response 5: Thank you for your suggestion. We have deleted the phrase “to our satisfaction” In Line 354 and modified as “The surgical removement combined with α-LNP administration significantly restricted the tumor growth and extended 20 days survival time, and displayed 1/8 completely remission Figure. 4C.” (Line 363-365)

Comments 6: Statistical analysis: The authors used parametric tests such as one-way ANOVA and Student’s t-test, which assume a normal distribution of the data. Clarification is needed on whether the normality of the data was assessed before applying these tests. Information on the normality tests conducted (e.g., Shapiro-Wilk test) and any transformations applied to meet the assumptions of parametric testing would be beneficial. If normality cannot be assumed, non-parametric alternatives (e.g., Kruskal-Wallis test, Mann-Whitney U test) should be considered and discussed.

Response 6: We apologize for the lack of clarity about the statistical analysis. We have checked the normality of the data by Normality test (Shapiro-Wilk test) by GraphPad Prism9.0 and the data were all normally distributed.

Major Concerns:

Comments 1: Vaccination schedule rationale: The rationale behind the vaccination schedule shown in Figure 2A is unclear. Typically, a priming and boost regimen involves a first immunization followed by a recall after at least 14 days to allow time for the adaptive immune response to develop, in a similar way to what performed in data associated to figure 3.

Response 1: The vaccination schedule of figure 2A was originally based on the references [1]. Both antigen-specific T-cell and humoral responses can be activated by mOVA-LNP in the vaccinated schedule [1, 2]. Similarly, we have also found that mRNA-LNP activated IFN-γ-spot-forming cells among splenocytes with this administrating strategy in previous study [3]. Therefore, we conducted the 5-day vaccinated schedule to detect the immune response of α-LNP.

1.             Miao L, Li L, Huang Y, Delcassian D, Chahal J, Han J, Shi Y, Sadtler K, Gao W, Lin J et al: Delivery of mRNA vaccines with heterocyclic lipids increases anti-tumor efficacy by STING-mediated immune cell activation. Nature biotechnology 2019, 37(10):1174-1185.

2.             Chen J, Ye Z, Huang C, Qiu M, Song D, Li Y, Xu Q: Lipid nanoparticle-mediated lymph node-targeting delivery of mRNA cancer vaccine elicits robust CD8(+) T cell response. Proceedings of the National Academy of Sciences of the United States of America 2022, 119(34):e2207841119.

3.             Xu Y, Hu Y, Xia H, Zhang S, Lei H, Yan B, Xiao ZX, Chen J, Pang J, Zha GF: Delivery of mRNA Vaccine with 1, 2-Diesters-Derived Lipids Elicits Fast Liver Clearance for Safe and Effective Cancer Immunotherapy. Adv Healthc Mater 2023:e2302691.

Comments 2: Assessment of humoral immunity: The authors evaluated the induction of specific antibodies after α-LNP vaccination (Figure 2A, B). However, the impact of humoral immunity in the antitumor preclinical evaluation has not been assessed either experimentally or in more details through discussion.

Response 2: Thank you for correction. we have done the supportive experiment related to specific antibodies. As Figure5 B is shown, we detect the GL7+ B cells and CD138+ plasmacytes on spleen, which related the antibody production. Besides, we have mentioned the induction of specific antibodies in discussion section (Line 448-452): “Humoral immunity resists the tumor through the direct neutralization and the antibody dependent cellular cytotoxicity (ADCC) effect. In our study, the significantly increased α-lactalbumin specific antibodies, GL7+ B cells and CD138+ plasmacytes have been observed in α-LNP vaccinated group mice, Therefore, we supposed that the anti-tumor efficiency might partly due to the ADCC response.”

Comments 3: Prophylactic potential and PD-1 therapy: The data describing the prophylactic and post-surgical rejection potential of α-LNP are encouraging, but the effect is not striking.

TNBC is generally considered a "cold" tumor, characterized by a lower level of immune activity within the tumor microenvironment, which often makes them less responsive to immunotherapy treatments compared to "hot" tumors that have higher levels of immune infiltration and activity. The interesting advantage of vaccination might lie in the enhancement of intratumoral T cell immunity, which is often kept silent in the immunotolerant tumor environment by PD-L1: PD-1 interaction. The presence of a higher number of T cells, especially CD8+ T cells, is a favorable prognostic factor. These “hot” tumors, characterized by substantial T cell infiltration, are more responsive to checkpoint inhibitor therapy.

Assessing the tumor microenvironment for T cell infiltration could provide critical insights.

·      Combination with Checkpoint Inhibitors: While the vaccination alone shows promising results, it might significantly enhance the impact of the work to combine it with anti-PD-1 therapy. This combination could help to reveal whether the vaccinated mice, despite displaying moderate control of tumor growth on their own, exhibit a more responsive and "hotter" tumor microenvironment due to increased T cell infiltration.

·      Experimental verification: This hypothesis could be verified by evaluating the immune cell infiltration status using techniques such as flow cytometry or immunofluorescence histology, similar to the methods employed in Figure 5. Specifically, assessing PD-1 expression on infiltrating CD8+ T cells and other immune markers would be beneficial.

·      Clinical implications: If the combination therapy proves effective, it could represent a significant advancement in the treatment of TNBC, making the tumors more responsive to checkpoint inhibitors and potentially transforming the therapeutic landscape for this aggressive cancer subtype.

·      Additional studies: Future studies should consider investigating the synergistic effects of α-LNP vaccination and PD-1 blockade in preclinical models. This approach could elucidate the potential for combined immunotherapy regimens to induce robust anti-tumor immunity and improve clinical outcomes in TNBC patients.

Response 3: Thank you for the constructive suggestion. We also hold a positive view on α-LNP combined PD-1 therapy might have a greater antitumor effect. From the immunofluorescence histology, we have found the PD-1 expresses 14.25% on infiltrating CD8+ T cells (figure in attachment word file), suggesting that PD-1 inhibitor might be enhance more 14.25% cytotoxicity to the 4T1-tumor, we will study the PD-1, CTLA-4 and other immune checkpoint inhibitors in detail in our future work.

Comments on the Quality of English Language

1.      English language: The manuscript could benefit from a thorough review to improve the overall English language quality.

Response1: Thanks for the suggestion. The expression of the manuscript has been improved with the help of a native English speaker. If necessary, we would like to have the professional improvement through the recommended service.

2.      Sentence structure: Authors frequently start sentences with “and,” which is grammatically incorrect in formal writing.

Response 2: Thanks for the suggestion. We have extensively scrutinized the sentences and deleted “And” in the start of sentence to be more formal.

(Line 321-323): “4T1 mammary carcinoma is a transplantable model is considered as a common TNBC model closed to the TNBC progress and metastases in clinic.”

(Line 332-334): “CD8+CD44+CD62L+ T cells (memory) and CD8+CD44+CD62L- T cells (effector) obviously showed an increase in infiltrating lymph node (Figure. 3E), suggesting that α-LNP developed an immunological memory response.”

(Line 50-51): “The protein primarily exists in TNBCs and these cell populations are positively correlated with the ‘triple-negative’ state and unfavorable prognosis of BC patients.”

(Line 464-465): “The α-LNP immunization of the model mice at an earlier tumor growth period displayed an improved antitumor effect, which is similar to the result of Vincent.”

Reviewer 2 Report

Comments and Suggestions for Authors

They have a logical proposal that α-lactalbumin mRNA-LNP may serve as a very promising target to TNBC through vaccination with mRNA neoantigens for a new therapeutic clinical strategy, based on the expression of α-Lactalbumin as a specific differentiation protein only expressed in mammary epithelial cells during lactation, and overexpressed in most of human breast cancer.

Since they dose not made yet the clinical trials, they should avoid to statement that they developed a new therapeutical strategy for TNBC, as a perspective it is okay and in the discussion as a preclinical model.

They mention Cell Culture of 4T1 (mouse breast cancer cell), Hela cell and HEK293T cell lines. What were Hela cells used for? In the respective figures and results, the type of cells involved should be noted.

It is necessary to write down the catalog numbers of the cells, the plasmid, MWCO cassette, etc.

It is necessary to check the spacing between words to ensure the readability of the page.

They must use same nomenclature in example (Fig. S1A) and Fig S1 (A); (Minipore, Cat # UFC500396) probably should be Millipore, and continuous order as in Figure 1D, then Figure S 1D, figure 1A-B, and so on. OR clearly annotated the molar ratio 44.2:9.9:44.2:1.7., maybe should be a molar ratio of 44.2:9.9 to 44.2:1.7.

Unless they are the owners, it is not necessary to use notations such as the INano™E (trademark) or Trans®.

If they immunized 5/6-week-old female Balb/C mice for 30 days, first immunized with 5 μg mRNA-LNP after 10 days, and then performed four more administrations every five days. How is that they use in the production of IFN-γ secreted by splenocytes, isolated 7 days after the last immunization, to measure by ELISPOT, bone marrow-derived dendritic cells collected from 6-8-week-old female Balb/C mice?

Furthermore, why is the immunization scheme with mRNA-LNP changed to three times for flow cytometry and sacrificed 7 days after the last administration? Differing from the scheme of immunotherapy.

Why they cite Figure S3 first than Figure S1? They must maintain a logical order.

Annotations such as: The modified mRNA vaccine, encoding α-lactalbumin, was encapsulated in LNPs. The α-lactalbumin mRNA was modified with N1-Methylpseudouridine-5′-Triphosphate(m1ψTP) to instead native UTP and then purified by RNA Clean magnetic Bead. It must be described in methods section.

They should set the length of their products as can be seen in Figure S 1D, in the capillary electrophoresis image of mRNA (α-RNA), α-LNP1 and α-LNP2, and refer to the ladders in the methods section. And what mean the bigger bands? Since they only refer that the western blots analysis revealed a clear band of 17 kD which suits the proper protein size of α-lactalbumin in results section. Note the meaning of the asterisks (****) somewhere in the text or in the figure legend, in example Figure 1E, if represents error bars mean ± SEM.

The authors should review the agreement between sections, for example they mention that the integrated form and the well-distributed state of α-LNP were evaluated by cryo-electron microscopy, but it does not appear in the methods section, nor do they should mix sections, for example, results with conclusions (Taken conclusion, with the well-evaluated parameters of α-LNPs, we successfully formulated the α-lactalbumin mRNA-LNP, and it could be expressed into α-lactalbumin well in vitro. Or in conclusion, our results revealed that the mRNA vaccine based on α-LNP have an adequate safety profile in vivo); Or results with discussion (These results demonstrated that α-LNP displayed a strong immunogenicity and showed a potential ability to activate anti-tumor immune reactions. However, robust immunogenicity of biologic drugs might be accompanied with immunity related problems including cytokines storm or other organs immune damage.).

References 1 and 2, these pages cannot be found. Sometimes they quote before the dot or after.

Comments on the Quality of English Language

Sometimes they quote references before the dot or after.

They must use same nomenclature in example (Fig. S1A) and Fig S1 (A).

(Minipore, Cat # UFC500396) probably should be Millipore.

It is necessary to check the spacing between words to ensure the readability of the page.

Author Response

Comments 1: Since they dose not made yet the clinical trials, they should avoid to statement that they developed a new therapeutical strategy for TNBC, as a perspective it is okay and in the discussion as a preclinical model.

Response 1: Thank you for pointing this out. We have modified the statement and discussed as a potential of therapeutical strategy for TNBC, as is followed.

(Line 19-20): “Here, we developed α-lactalbumin mRNA-lipid nanoparticles (α-LNP) as a potential therapeutical strategy for TNBC.”

(Line 79): “In this work, we proposed a potential clinical strategy for TNBC therapeutics.”

Comments 2: They mention Cell Culture of 4T1 (mouse breast cancer cell), Hela cell and HEK293T cell lines. What were Hela cells used for? In the respective figures and results, the type of cells involved should be noted

Response 2: We apologize for the lack of clarity in our original statement. The strains Hela cell was used to transfection experiment of α-lactalbumin mRNA. While the expression of the mRNA in Hela cell cannot be detected by western blotting. We have now deleted “Hela cell” in method section.

(Line 88-89): “4T1 mouse breast cancer cell (CAT#CL-0007) and HEK293T cell lines (CAT#CL-0005) were purchased from Procell Life Science &Technology Co, Ltd.”

Comments 3: It is necessary to write down the catalog numbers of the cells, the plasmid, MWCO cassette, etc.

Response 3: Thank you for pointing this out. We have now added the information of catalog numbers of the cells, the plasmid, MWCO cassette, etc. in the revised manuscript.

(Line 88-89): “4T1 mouse breast cancer cell (CAT#CL-0007) and HEK293T cell lines (CAT#CL-0005) were purchased from Procell Life Science &Technology Co, Ltd.”

(Line 94-97): “The mRNA expression plasmids (pIVT), which is reconstructed based on pUC57 (GenScript, CAT#1176), contained T7 promoter, the optimized protein-coding sequence…”

(Line 128-130): “Then the mRNA-LNP mixture were diluted with 1X PBS in a 100,000 MWCO cassette (Merck, CAT#UFC9100) at 4 °C for 30 min for three time, and was centrifuged at 1200 g for 30 min, then stored at 4 °C before use.”

Comments 4: It is necessary to check the spacing between words to ensure the readability of the page.

Response 4: Thank you for pointing this out. We have checked the spacing between words in the whole manuscript and revised it.

Comments 5: They must use same nomenclature in example (Fig. S1A) and Fig S1 (A); (Minipore, Cat # UFC500396) probably should be Millipore, and continuous order as in Figure 1D, then Figure S 1D, figure 1A-B, and so on. OR clearly annotated the molar ratio 44.2:9.9:44.2:1.7., maybe should be a molar ratio of 44.2:9.9 to 44.2:1.7.

Response 5: We have now corrected the order of the figures and used the same nomenclature in the revised manuscript.

We apologize for the lack of clarity in our original statement. And we have modified the original statement “The lipids included the synthesized ionizable lipid, DSPC (Avanti), cholesterol (Avanti) and DMG-PEG 2000 (Avanti), solubilized by ethanol with a molar ratio of 44.2:9.9:44.2:1.7.” to “The lipids included the synthesized ionizable lipid, DSPC (Avanti), cholesterol (Avanti) and DMG-PEG 2000 (Avanti), with a molar ratio of 44.2:9.9:44.2:1.7, which are solubilized by ethanol.” (Line 125-127)

Comments 6: Unless they are the owners, it is not necessary to use notations such as the INano™E (trademark) or Trans®.

Response 6: Thank you for pointing this out. We have now deleted trademark of INano™E and Trans®.

(Line 124-125): “Lipids and mRNA were mixed in a microfluidic device (MicroNano) at a 3:1 volume ratio (N/P ratio of lipids/mRNA was 5:1).”

(Line 139): “The 24-well plate was inoculated with 2.5×105 HEK293T cells for each well 24 hours be-fore the transfection of 0.5 μg mRNA per well using Liposomal Transfection Reagent (YEASEN, Cat # 40802ES03).”

Comments 7: If they immunized 5/6-week-old female Balb/C mice for 30 days, first immunized with 5 μg mRNA-LNP after 10 days, and then performed four more administrations every five days. How is that they use in the production of IFN-γ secreted by splenocytes, isolated 7 days after the last immunization, to measure by ELISPOT, bone marrow-derived dendritic cells collected from 6-8-week-old female Balb/C mice?

Response 7: We apologize for the lack of clarity in our original statement. We had used different mice to detect the ELISPOT experiment and 5-times-administrating experiment, which were performed with same dose of administrations.

We have used 5 mice per group to do the ELISPOT experiment: we detected the production of IFN-γ secreted by splenocytes, isolated from 3-times-administrating mice. For 5-times-immunotherapy, 24 mice (8 mice/group) were performed to analyze the antitumor effect of mRNA-LNP. Considering the same quality and administrating dose of mRNA-LNP and the same origin of mice, we think the mice had the similar stereotype or stronger IFN-γ-secreting splenocytes in 5-times-administrating experiment, compared to the 3-times-administrating mice. As for the bone marrow-derived dendritic cells, these bone marrow cells were collected from a new-batch Balb/C mice.

Comments 8: Furthermore, why is the immunization scheme with mRNA-LNP changed to three times for flow cytometry and sacrificed 7 days after the last administration? Differing from the scheme of immunotherapy.

Response 8: Thank you for correction. We had used different immunization scheme to detect the antitumor response of α-LNP. Mice with 3-times-immunization were used to detect the antitumor effect of α-LNP before tumor implantation (tumor-prophylactic effect), then we detect the immune activation in lymph nodes and spleen by flow cytometry in Figure 3. The 3-times-immunization strategy is based on the antitumor study of B16-OVA melanoma[1, 2].In Figure 4-5, we immunized the mice for five times to detect the antitumor effect of α-LNP after tumor implantation (tumor-therapeutical effect), then detect the immune responses by flow cytometry, etc. Here, we adjust the administration to 5 times for gaining the better antitumor effect. In our result, α-LNP elicited the antitumor responses, so we immunized for the whole tumor-bearing time.     

Comments 9: Why they cite Figure S3 first than Figure S1? They must maintain a logical order.

Response 9: Thank you for correction. We have now modified the order of the figures in logic in the revised manuscript.

Comments 10: Annotations such as: The modified mRNA vaccine, encoding α-lactalbumin, was encapsulated in LNPs. The α-lactalbumin mRNA was modified with N1-Methylpseudouridine-5′-Triphosphate(m1ψTP) to instead native UTP and then purified by RNA Clean magnetic Bead. It must be described in methods section.

Response 10: We apologize for the neglection and have now added the details of RNA purification in methods section in the revised manuscript (Line 112-117).

Methods: RNA Clean Beads was added to original RNA solution and mix well. Incubation for 5 min at room temperature to make the RNA-binding on magnetic beads. Then sample placed in a magnetic rack for 5 min, carefully remove the supernatant. Keep the tube in the magnetic rack, and add in 500 μL 80% ethanol to wash magnetic beads, incubating for 30 sec, carefully remove the supernatant and repeat the wash step once. Eluting and collect RNA from the magnetic rack by adding suitable volume of H2O Nuclease-free.

Comments 11: They should set the length of their products as can be seen in Figure S 1D, in the capillary electrophoresis image of mRNA (α-RNA), α-LNP1 and α-LNP2, and refer to the ladders in the methods section. And what mean the bigger bands? Since they only refer that the western blots analysis revealed a clear band of 17 kD which suits the proper protein size of α-lactalbumin in results section. Note the meaning of the asterisks (****) somewhere in the text or in the figure legend, in example Figure 1E, if represents error bars mean ± SEM.

Response 11: We apologize for the low quality of Figure S 1D and we have added the marker ladder in the revised figure in attachment.

The bigger bands in Figure S 1D may be the combination of mRNA (α-RNA) and ionizable lipid of LNP. Similarly, Meredith’s paper also found a bigger RNA peak in mRNA-LNP sample, as can been seen in figure[3](please check the attachment). The mRNA (α-RNA) was released from LNP in the capillary electrophoresis experiment and may bond to the cationic lipids (synthetic ionizable lipids) of LNP, which form bands with larger molecular weight.

Besides, we have now completed the meaning of sterisks (****) in the figure1 legend.

“The quantitation data was represented with 3 experimental replicates, *p < 0.05, **p < 0.01, ***p < 0.001, ****p < 0.0001, error bars represent mean ± SEM.” (Line 280-282)

Comments 12: 11. The authors should review the agreement between sections, for example they mention that the integrated form and the well-distributed state of α-LNP were evaluated by cryo-electron microscopy, but it does not appear in the methods section, nor do they should mix sections, for example, results with conclusions (Taken conclusion, with the well-evaluated parameters of α-LNPs, we successfully formulated the α-lactalbumin mRNA-LNP, and it could be expressed into α-lactalbumin well in vitro. Or in conclusion, our results revealed that the mRNA vaccine based on α-LNP have an adequate safety profile in vivo); Or results with discussion (These results demonstrated that α-LNP displayed a strong immunogenicity and showed a potential ability to activate anti-tumor immune reactions. However, robust immunogenicity of biologic drugs might be accompanied with immunity related problems including cytokines storm or other organs immune damage.).

Response 12: We have now added the details of cryo-electron microscopy in methods section.

(Line 156-160) Cryo-electron microscopy: The LNP sample in solution was spread on a grid (Xinxing ProMed, CAT#T11032, #T11012), forming a very thin liquid. Then the grid is put into ethane for flash freezing. After flash freezing, the sample will be transformed into amorphous ice. Cryo-EM Sample are prepared by the vitrobot cryoprotographer (Thermo Fisher) and then observed by TEM (Kiros G4, Einstein).

Besides, we have modified the sentences in the section of result and conclusion.

(Line 266-268): “With the well-evaluated parameters of α-LNPs, we successfully formulated the α-lactalbumin mRNA-LNP, and it could be expressed into α-lactalbumin well in vitro.”

(Line 301-303): “Our results revealed that the mRNA vaccine based on α-LNP have an adequate safety pro-file in vivo. Collectively, α-LNP is a promising TNBC mRNA vaccine candidate for its the strong immunogenicity and safety profile in vivo.”

(Line 338-339): “Based on our results, α-LNP initiated effective anti-tumor immunity and prevent the growth of 4T1-transplantable TNBC tumor model.”

(Line 291-294): “To evaluate the cytokines in blood, we then detected the level of interferon-γ (IFN-γ), tumor necrosis factor-α (TNF-α) and interleukin-12 (IL-12) in peripheral blood. After three treatments of α-LNP with an interval of 5 days, the level of IFN-γ, TNF-α and IL-12 of experiment mice showed no obvious change in serum.”

(Line 460-463): According to previous studies, the tumor burden-mice conducted surgical removement displayed rapid recurrence, we also found that single surgery remove 4T1 relapsed quickly, while the surgery combined with α-LNP administration significantly restricted the tumor growth and suppressed the lung metastasis of TNBC.

Comments 13: References 1 and 2, these pages cannot be found. Sometimes they quote before the dot or after.

Response 13: We The pages of reference 1 and 2 can be opened by clicking the website links. Please check again:

https://www.who.int/news/item/03-02-2021-breast-cancer-now-most-common-form-of-cancer-who-taking-action

https://www.who.int/news-room/fact-sheets/detail/breast-cancer

 We have now checked out the whole manuscript and kept the reference quotation before the dot. “Moreover, some studies indicate that women have a substantial proinflammatory T cell repertoire responding to human recombinant α-lactalbumin [5].” (Line53)

Comments on the Quality of English Language

Comments 1: Sometimes they quote references before the dot or after.

Response 1: We have now checked out the whole manuscript and kept the reference quotation before the dot.

(Line 53): “Moreover, some studies indicate that women have a substantial proinflammatory T cell repertoire responding to human recombinant α-lactalbumin [5].

Comments 2: They must use same nomenclature in example (Fig. S1A) and Fig S1 (A).

Response 2: We have now checked out the whole manuscript and kept all in same nomenclature of (Fig. S1A)

Comments 3: (Minipore, Cat # UFC500396) probably should be Millipore.

Response 3: Thank you for correction. The “Minipore” have been corrected to “Millipore” in the revised manuscript. (Line 142)

Comments 4: It is necessary to check the spacing between words to ensure the readability of the page.

Response 4: Thank you for the suggestion. We have checked the spacing between words in the whole manuscript and revised it. And our manuscript is based on the Microsoft Word template (https://www.mdpi.com/journal/pharmaceutics/instructions).

Reviewer 3 Report

Comments and Suggestions for Authors

In “α-lactalbumin mRNA-LNP evokes anti-tumor effect combined with surgery in Triple Negative Breast Cancer” Yun-Ru He and col. Investigate the effect of α-lactalbumin mRNA-LNP (α-LNP) vaccine for both therapy and prophylactics in triple negative breast cancer (TNBC). Importantly, the authors performed in vivo experiments to evaluate the safety of α-LNP.

In particular, they found that α-LNP activates cellular and humoral anti-tumor immunity and in combination with basic surgery displays a favourable therapeutic response on the murine TNBC model by extending survival time and reducing recurrence. Also, they found that vaccination previous to tumor implantation has a preventive effect against tumor growth and extended the survival of mice with 4T1 TNBC tumors.

This work brings a new therapeutic strategy for TNBC, the subtype of breast cancer that still lack an effective clinical treatment yet. I consider this work should be published with minor revisions.

Minor comments:

1.       In figure 1, subfigures should be placed according are described in the manuscript, or conversely, the subfigures should be described according were ordered in the figure.

2.       In figure 3 the legend of figure 3D correspond to figure E upper, and the legend for Figure 3D is missing.

3.       In Figure 4D, “volumn” should be replaced by “volume”.

4.       The original Gel from Figure 1D seems not to be the same showed in the manuscript (the result is the same).

5.       In Materials and Methods, when describe statistical analyses using Microsoft Excel and Prism 9.0 (GraphPad); since the authors performed parametrical analyses how the authors check equal variance beside normality?

6.       In Materials and Methods section, when describe the synthesis of α-Lactalbumin, authors should describe the source of genomic DNA used to amplify this mRNA.

7.       Along the manuscript, numbers in molecular formulas should have the numbers in subscript position (for example line 92, CO2). Also values ​​expressed in scientific notation must have powers in superscript (for example number of cells injected to the mice).

8.       In Materials and methods the place of administration of α-LNP (intramuscular according to results methods) should be described.

9.       In line 198 “Flow Cytometory” should be replaced by “Flow Cytometry”.

10.   In line 456, “reoccurrence” should be replaced by “recurrence”.

Comments on the Quality of English Language

There are minimal errors that I indicated as minor comments above.

Author Response

Comment 1:In figure 1, subfigures should be placed according are described in the manuscript, or conversely, the subfigures should be described according were ordered in the figure.

Response1: We have now revised the order logic of the Figure1 and the figure legend (Line 271-274) as described in the manuscript.

(Line 227): “The flow cytometry strategies were illustrated in figure S1.”

(Line 258-259): “high performance capillary electrophoresis results also showed the mRNA with good condition before and after encapsulation (Figure. S2).”

(Line 298-300): “In addition, tissues of main organs from the experiment-grouped mice were isolated for histopathological examination, and there were no pathological changes between the vaccination and control groups (Figure. S3).”

Comment 2:In figure 3 the legend of figure 3D correspond to figure E upper, and the legend for Figure 3D is missing.

Response 2: We have revised the legend of Figure 2F and deleted “(F)The serum level of ALT/AST/BUN/CR in different groups” (Line 315-316).

Comment 3:In Figure 4D, “volumn” should be replaced by “volume”.

Response 3: Thank you for correction. The “volumn” in figure 4D have been corrected to “volume” in the revised manuscript.

Comment 4:The original Gel from Figure 1D seems not to be the same showed in the manuscript (the result is the same).

Response 4: Thank you so much for correction. We apologize that the original gel was mispacked in the supplementary folder. The original gel from Figure 1D is followed in attachment.

Comment 5:In Materials and Methods, when describe statistical analyses using Microsoft Excel and Prism 9.0 (GraphPad); since the authors performed parametrical analyses how the authors check equal variance beside normality?

Response 5: We apologize for the lack of clarity about the statistical analysis. We have checked the normality of the data by Normality test (Shapiro-Wilk test) by Prism 9.0 (GraphPad) to ensure the data were all normally distributed.

Comment 6:In Materials and Methods section, when describe the synthesis of α-Lactalbumin, authors should describe the source of genomic DNA used to amplify this mRNA.

Response 6: The α-lactalbumin ( Gene NP_034809.1) was optimized and inserted into IVT plasmid for mRNA synthesis, optimized CDS sequence (bottom-lined ) is followed in the attachmeng.

Comment 7:Along the manuscript, numbers in molecular formulas should have the numbers in subscript position (for example line 92, CO2). Also values ​​expressed in scientific notation must have powers in superscript (for example number of cells injected to the mice).

Response 7: Thank you for correction. We have the numbers in scientific position.

(Line 91-92): “All cells were grown at 37 °C in 5% CO2 conditions.”

(Line 170-171): “For immunotherapy model, 2×105 4T1 cells were injected subcutaneously in the right flank of the Balb/C mice.”

(Line 174): “Then 2×105 4T1 cells were injected subcutaneously in the right flank of the Balb/C mice.”

(Line 204-207): “When CD11c express in over 70% of cells, the BMDCs were incubated with 1 μg mRNA-LNP in 6-well plate (4×105 cells per well) for 24h. The splenocytes (2×105 cells per well) from the immunized mice were harvested and co-incubated with the immunized BMDCs (5×104 cells per well) at ratio of 4:1 in ELISPOT plate.”

(Line 175-176): “Tumor volume was measured with formulation of ½ × (length × width × height) every 2/3 days and the tumors were limited by a volume of 1500 cm3.”

Comment 8:In Materials and methods the place of administration of α-LNP (intramuscular according to results methods) should be described.

Response 8: Thank you for correction. We have now stated the way of administration, as show as (Line 173) “the mice were intramuscularly immunized with 3 μg α-LNP or LUC-mRNA-LNP or non-RNA-LNP or PBS for three time every 7 days.”

Comment 9:In line 198 “Flow Cytometory” should be replaced by “Flow Cytometry”.

Response 9: Thank you for correction. The “Flow Cytometory” in Line 210 have been corrected to “Flow Cytometry” in the revised manuscript.

Comment 10:In line 456, “reoccurrence” should be replaced by “recurrence”.

Response 10: Thank you for correction. The “reoccurrence” in Line 463 have been corrected to “recurrence” in the revised manuscript.

Round 2

Reviewer 1 Report

Comments and Suggestions for Authors

The Authors have addressed all of my concerns with the original manuscript.

Author Response

Thank you very much for your thorough review and for approving our manuscript.

Reviewer 2 Report

Comments and Suggestions for Authors

The authors reviewed and addressed each of the observations throughout the entire manuscript, so it is recommended to accept it.

If I click the link for the pages of reference 1 and 2 still sending error 404. But if I copy and paste, it works. Please check. No further review is necessary

Author Response

(The authors gave the same response as above.)

Reviewer 3 Report

Comments and Suggestions for Authors

This is the second revision of “α-lactalbumin mRNA-LNP evokes anti-tumor effect combined with surgery in Triple Negative Breast Cancer”.

Authors have answered almost all issues I have suggested but still there two things to improve.

I consider that this paper should be accepted with minor revisions:

1. In Materials and Methods, when describe statistical analyses authors still do not describe how they have checked equal variance, that is a requisite to perform parametric test beside normality.

2. To mection Figure 3D in the manuscript, in line 326 where say “11 survival-days (Figure. 3A-C)” should say “11 survival-days (Figure. 3A-D)”.

Author Response

Comments 1: In Materials and Methods, when describe statistical analyses authors still do not describe how they have checked equal variance, that is a requisite to perform parametric test beside normality.

Response 1: We apologize for the lack of clarity about the statistical analysis. We have checked the normality of the data by Normality test (Shapiro-Wilk test). Before performing parametric tests, we used Levene’s test to assess the homogeneity of variances. The test evaluates whether the variances across different groups are equal, thus ensuring that the prerequisites for parametric tests are met. Besides, we have completed the statement in the statistical analyses section. 

 (Line 242-243): “Normality test was checked by Shapiro-Wilk test and equal variance was checked by Levene’s test.”

Comments 2: To mention Figure 3D in the manuscript, in line 326 where say “11 survival-days (Figure. 3A-C)” should say “11 survival-days (Figure. 3A-D)”.

Response 2: Thank you for pointing this out. We have corrected the “11 survival-days (Figure. 3A-C)” should say “11 survival-days (Figure. 3A-D)” in Line 326.

(Line 326): “In the 4T1 xenograft mice model, we observed significant inhibitory effect on tumor growth and a total extension of 11 survival-days (Figure. 3A-D) in α-LNP treated group mice.”
